

# A new Amazonian species of *Allobates* Zimmermann & Zimmermann, 1988 (Aromobatidae) with a trilled advertisement call

Leandro A. Silva[1], Ricardo Marques[1], Henrique Folly[2] and Diego J. Santana[2]

[1] Centro de Ciências Exatas e da Natureza, Universidade Federal da Paraíba, João Pessoa, Paraíba, Brazil
[2] Instituto de Biociências, Universidade Federal de Mato Grosso do Sul, Campo Grande, Mato Grosso do Sul, Brazil

## ABSTRACT

**Background:** Currently, 58 species are assigned to the genus *Allobates*, with 70% of its diversity described just in the last two decades, with many additional species likely unnamed. The continuous description of these new species represents a fundamental step for resolving the taxonomy and ensuring the future conservation of the genus.
**Methods:** Using molecular, acoustic, and morphological evidences, we describe a new species of *Allobates* from Teles Pires River region, southern Amazonia, and provide accounts on the population of *A. tapajos* found sympatrically with the new species.
**Results:** The new species is distinguished from its congeners by the coloration of thighs, venter, dorsum, and dark lateral stripe. It has four types of calls, with advertisement calls formed by relatively long trills with a mean duration of 2.29 s ± 0.65, mean of 39.93 notes ± 11.18 emitted at a mean rate of 17.49 ± 0.68 notes per second, and mean dominant frequency of 5,717 Hz ± 220.81. The genetic distance between the new species and its congeners in a fragment of the 16S mitochondrial fragment ranged between 13.2% (*A. carajas*) to 21.3% (*A. niputidea*). The sympatric *Allobates* population fits its morphology and acoustic with the nominal *A. tapajos*, but presents a relatively high genetic distance of nearly 6.5%, raising questions on the current taxonomy of this species.

## INTRODUCTION

The number of species of *Allobates* Zimmermann & Zimmermann, 1988 has continuously increased during the last two decades (*e.g. Simões et al., 2018*; *Moraes, Pavan & Lima, 2019*; *Simões, Rojas & Lima, 2019*; *Souza et al., 2020*; *Jaramillo et al., 2021*), but several phenotypically and molecularly lineages remain unnamed (see *Simões, Lima & Farias, 2010*; *Grant et al., 2017*; *Fouquet, Vidal & Dewynter, 2019*; *Lima, Ferrão & Silva, 2020*; *Réjaud et al., 2020*). Currently, the genus includes 58 species allocated into four groups: the

Corresponding author
Leandro A. Silva,
leandroherpeto@dse.ufpb.br

Atlantic Forest group [one species, *A. olfersioides* (*Lutz, 1925*)], the trans-Andean group [two species, *A. niputidea* Grant, Acosta & Rada, 2007 and *A. talamancae* (*Cope, 1875*)], the colorful *A. femoralis* group [four species, *A. femoralis* (*Boulenger, 1884*), *A. hodli* *Simões, Lima & Farias, 2010*, *A. myersi* (*Pyburn, 1981*), and *A. zaparo* (*Silverstone, 1976*)], and the most diverse group that includes 51 valid species [*e.g. A. brunneus* (*Cope, 1887*), *A. carajas* *Simões, Rojas & Lima, 2019*, *A. crombiei* (*Morales, 2002*), *A. grillisimilis* *Simões et al., 2013a*, and *A. tapajos* *Lima, Simões & Kaefer, 2015* (*Grant et al., 2017*)].

The Brazilian Amazonia harbors most of the *Allobates* diversity, with 30 species occurring in this tropical rainforest (*e.g. Frost, 2021*; *Segalla et al., 2021*). The origin of the current genus-level diversity likely emerged through vicariant speciation resulting from an interplay between Miocene events, mainly during the Pebas System between 14 and 10 Mya, and secondarily with the comparatively more recent establishment of the major Amazonian rivers (*Réjaud et al., 2020*). In addition to the aforementioned impacts of allopatry, variation in vegetation, altitude, and geomorphology likely influenced the current species distribution and even speciation events (*Lima, Ferrão & Silva, 2020*). Undoubtedly, the accelerated description of new species of these charismatic nurse-frogs have greatly improved our knowledge on the distribution of species richness within the genus, and shed light on the processes that generated such extensive species diversity.

Recent studies have highlighted the conservative morphology of *Allobates* species (*e.g. Carvalho, Martins & Giaretta, 2016*; *Moraes & Lima, 2021*), and taxonomic decisions considering an integrated systematic approach are therefore crucial to fully describing the diversity within the genus (*Carvalho, Martins & Giaretta, 2016*; *Grant et al., 2017*; *Simões et al., 2018*; *Moraes, Pavan & Lima, 2019*; *Simões, Rojas & Lima, 2019*). This effort is especially important for diagnosing sympatric lineages, which is the case for several Amazonian *Allobates* species descriptions (*Simões, Lima & Farias, 2010*; *Lima, Ferrão & Silva, 2020*; *Moraes & Lima, 2021*).

During fieldworks between 2015 and 2019 along the banks of the Teles Pires River, southern Amazonia (Fig. 1), we found two sympatric and cryptically colored *Allobates* lineages. In order to assess their taxonomic status, we analyzed their morphology, advertisement calls, and fragments of 16S mitochondrial gene. These data identified one to be a previously reported genetically divergent lineage of *A. tapajos* (*A.* aff. *tapajos* 3, *sensu Réjaud et al., 2020*), whereas the second represents a lineage previously unknown to science, which is also phenotypically and acoustically diagnoseable from its congeners. Here we describe the latter lineage as a new species of *Allobates*. We also provide brief accounts on the sympatric *Allobates* population.

# MATERIAL AND METHODS

## Study area and sampling

We performed field work on both banks of the Teles Pires River, in Jacareacanga municipality, southern Pará state, and in Paranaíta municipality, northern Mato Grosso state, Brazil, between November 2015 and November 2019 (Fig. 1). During diurnal surveys at different sites, we collected 25 specimens of *Allobates*, 10 representing the new species described below and 15 representing the *A. tapajos* lineage. We euthanized specimens

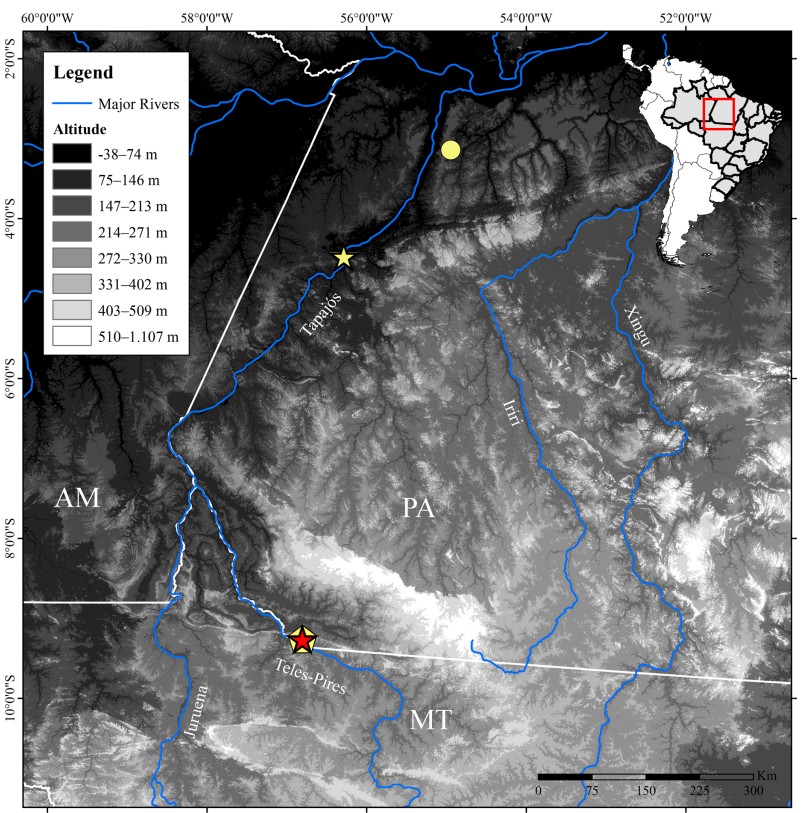

**Figure 1 Geographic distribution of *Allobates paleci* sp. nov. and *A. tapajos* in Brazilian Amazonia.** Stars: type localities of *A. paleci* sp. nov. (red, Jacareacanga-PA) and *A. tapajos* (yellow, Itaituba-PA); Circle: a second record of *A. tapajos* for Santarém-PA. Pentagon: lineage of *A. tapajos* from Paranaíta-MT. Abbreviations: AM, Amazonas state; MT, Mato Grosso State; PA, Pará state.

using a liquid solution of 2% lidocaine chlorhydrate, preserved them in 10% formalin, and posteriorly stored them in 70% ethanol. Tissue samples from muscles were taken before specimen preservation and stored in 100% ethanol. Collection permits were issued by Instituto Chico Mendes de Conservação da Biodiversidade (ICMBio/SISBIO #79127-1). Specimens were accessioned in the Coleção Zoológica da Universidade Federal de Mato Grosso do Sul (ZUFMS-AMP), Campo Grande, Mato Grosso do Sul, and in the Coleção Herpetológica da Universidade Federal da Paraíba, João Pessoa, Paraíba (UFPB). Analyzed specimens are listed in Appendix 1.

## Bioacoustics analysis

We recorded calls from nine males of the new species from Jacareacanga, and eight males of *Allobates* aff. *tapajos*, seven from Jacareacanga and one from Paranaíta. All calls were recorded using a Tascam DR-40 digital recorder with built in microphones at 44.1 kHz with a 16-bit resolution. The recordings were made between 7:30 and 18:00 h, air temperature range was 24–30 °C and air humidity 80–90%. We analyzed calls in Raven Pro v. 1.5 (*Center for Conservation Bioacoustics, 2014*). Temporal parameters were measured from oscillograms, and spectral parameters were measured from spectrograms.

Remaining sets were: Hann window type, FFT size = 256, brightness 67%, and contrast 70%. To reduce background noise we applied a 2,500-Hz high-pass filter before acoustic analyses.

We describe four call types for the new species: (i) calls composed of single notes (voucher CHUFPB30245 (field number: AAGARDA12596), three calls); (ii) warming-up short calls (3–15 notes; nine males, 72 calls and 216 notes); (iii) advertisement calls (17–61 notes; nine males, 61 calls and 549 notes); and (iv) singular multipulsed notes (16–33 pulses; voucher CHUFPB30256 (AAGARDA12595), six calls). For single notes calls, we measured the following temporal and spectral parameters: call duration, silent interval between calls and dominant, minimum, and maximum frequencies of the call. Because warming-up and advertisement calls presented a multi-note structure, we sampled a subset of notes to describe call's temporal and spectral parameters. For warming-up calls, we measured the following parameters of the whole call and the first, most central, and last notes of each call (*i.e. n* = 3 notes per call): Temporal parameters – call duration, silent interval between advertisement calls, number of notes per call, duration of the first three notes, silent interval between the first three notes, duration of the most central three notes, silent interval between the three most central notes, duration of last three notes, silent interval between the last three notes, and note repetition rate. Spectral parameters – dominant, minimum and maximum frequencies of the whole call, first three notes, most central three notes, and last three notes.

For advertisement calls, we measured the following parameters of the whole call and the first three, three most central, and last three notes of each call (*i.e. n* = 9 notes per call). Temporal parameters – call duration, silent interval between the advertisement calls, number of notes per call, duration of first three notes, silent interval between the first three notes, duration of the most central three notes, silent interval between the three most central notes, duration of last three notes, silent interval between the three last notes, and note repetition rate. Spectral parameters – dominant, minimum and maximum frequencies of the whole call, first three notes, most central three notes, and last three notes. For calls composed of multipulsed singular notes, we measured the call duration, silent interval between the multipulsed singular notes, number of pulses per call, and dominant, minimum and maximum frequencies of each call. For calls consisting in single notes, we measured call duration, silent interval between calls, and dominant, minimum and maximum frequencies of the call.

Because the sympatric *Allobates tapajos* presented a different call structure from the new species, we adopted the methodology proposed by *Lima, Simões & Kaefer (2015)* for its call description with minor adaptations. From each recording, we selected a section with uninterrupted calls around the middle length of the recording. In this section we analyzed 20–24 calls and 34–52 notes per recording, from which we measured both temporal and spectral parameters: duration, dominant, minimum and maximum frequencies, silent interval between the calls, and silent interval between notes in all recordings. We also estimated the rate of note emission (number of notes/seconds) from these sections in all recordings.

## Morphology

We measured specimens using a digital caliper to the nearest 0.1 mm. We followed *Fabrezi & Alberch (1996)*, *Grant et al. (2006)*, *Lima, Sanchez & Souza (2007)*, and *Barrio-Amorós & Santos (2009)* for morphometric measurements and terminology: snout-to-vent length, head length from tip of snout to posterior edge of maxilla articulation, head width at the level of maxilla articulation, snout length from tip of snout to the center of nostril, eye-to-nostril distance from anterior corner of the eye to the center of nostril, internarial distance, eye diameter from anterior to posterior corner, interorbital distance, maximum diameter of tympanum, forearm length from proximal edge of palmar tubercle to outer edge of flexed elbow, upper arm length from anterior corner of arm insertion to the outer edge of flexed elbow, lengths from proximal edge of palmar tubercle to tips of fingers II, III, IV and V; width of disc on finger III, width of finger III's third phalanx, diameter of palmar tubercle, diameter of the nar tubercle, leg length from the posterior extremity of the urostyle region to the outer edge of flexed knee, tibia length from outer edge of flexed knee to heel, foot length from proximal edge of outer metatarsal tubercle to tip of toe IV, and width of disc on toe IV (WTD).

## Molecular analysis

We extracted DNA from tissue samples using the sodium-chloride salt precipitation method (*Bruford et al., 1992*). For the polymerase chain reaction (PCR) amplification, we used 7.5 µl of Taq DNA Polymerase Master Mix (Ampliqon S/A, Denmark), 0.4 µl of either primer (forward/backward), and 1–2 µl of DNA, then we complemented with Milli-Q water for a final volume reaction of 15 µl. Then, we amplified a fragment of the mitochondrial DNA gene 16S using primers 16Sar and 16Sbr of *Palumbi (1996)*. The PCR protocol was configured with one initial cycle of 94 °C for 3 min, followed by 35 cycles of 94 °C for 20 s, 48 °C for 20 s, 68 °C for 40 s, and a final extension cycle of 68 °C for 5 min. The purification of PCR products and sequencing were performed by Macrogen Inc. (Seoul, South Korea).

We compared the newly generated 16S sequences with all *Allobates* sequences with compatible fragment region deposited in GenBank, including 43 nominal species and several unassigned sequences. Due to the immense amount of 16S sequences available (>850 sequences), many of them from the same species, we chose, when available, up to three sequences from each species and candidate/non-described species identified in previous studies (*e.g. Simões, Lima & Farias, 2010*; *Grant et al., 2017*; *Fouquet, Vidal & Dewynter, 2019*; *Lima, Ferrão & Silva, 2020*). We aligned the 16S mtDNA gene fragments using MAFFT algorithm (*Katoh et al., 2002*) in Geneious v 9.0.5 with default settings. The final dataset comprised 186 sequences of a 387 base pairs (bp) fragment of the 16S gene (Document S1). All GenBank accession numbers and genetic vouchers used here are listed in the Table S1. We implemented a Maximum Likelihood tree inferred in RAxML (*Stamatakis, 2014*) *via* raxmlGUI 2.0 (*Edler et al., 2021*). We run the analysis using a ML + rapid bootstrap setting with TIM2+I+G substitution model and 1,000 bootstrap replicates. Substitution model was tested with Modeltest (*Darriba et al., 2020*) in raxmlGUI 2.0. In addition, we ran a PTP species delimitation analysis (*Zhang et al., 2013*)

using the ML Tree, which the calculations were performed on PTP websever (http://species.h-its.org/ptp/), with 500,000 MCMC generations, thinning set at 100 and burn-in at 10%. We also ran a Bayesian Inference (BI) performed in BEAST v.2.6.3 (*Bouckaert et al., 2019*) for 20 million generations, sampling every 2,000 steps using a Yule Process tree prior. We checked for stationarity by visually inspecting trace plots and ensuring that all values for effective sample size were above 200 in Tracer v1.7.1 (*Rambaut et al., 2018*). The first 10% of sampled genealogies were discarded as burn-in, and the maximum clade credibility tree with median node ages was calculated with TreeAnnotator v.2.6.3 (*Bouckaert et al., 2019*). With this tree, we ran a Generalized Mixed Yule Coalescent (GMYC) for species delimitation (*Pons et al., 2006*; *Fujisawa & Barraclough, 2013*) in the R v 4.1.1 (*R Core Team, 2021*) by using the package *splits* (*Ezard, Fujisawa & Barraclough, 2017*). We calculated sequence divergences (uncorrected p-distances) among species/individuals using MEGA v10.1.1 (*Kumar et al., 2018*).

## Nomenclatural acts

The electronic edition of this article conforms to the requirements of the amended International Code of Zoological Nomenclature, and hence the new names contained herein are available under that Code of this article. This published work and the nomenclatural acts it contains have been registered in ZooBank, the online registration system for the ICZN. The LSID (Life Science Identifier) for this publication is: urn:lsid:zoobank.org:pub:6B0FDB3B-30B2-471E-9E06-8B7F296F454F. The electronic edition of this work was published in a journal with an ISSN, has been archived, and is available from the following digital repository: www.peerj.com/.

# RESULTS

## Morphology

The new species is cryptically colored and has an hourglass mark on its dorsum, being easily distinguished from most of the Brazilian congeners. The most similar congener is *A. tapajos*, from which the new species can be diagnosed by presents an interrupted dark lateral stripe, more conspicuous anteriorly (dark lateral stripe gradually fades towards the inguinal region in *A. tapajos*). As detailed below, however, these two species are acoustically and molecularly easily diagnosable. The overall coloration of the *A. tapajos* population from the Teles Pires River consists of a light brown dorsum with irregular dark brown blotches, ventral surfaces of males golden yellow on throat and chest, and white to yellow on abdomen, an identical pattern to that indicated in the original description (*Lima, Simões & Kaefer, 2015*).

## Bioacoustics

The acoustic repertoire of the new species includes four call types: advertisement, warmup, single unpulsed notes, and single multipulsed notes. The advertisement call of the new species has a multi-note structure with a high note repetition rate (0.97–3.57 s, 17–61 notes, 16.3–19.1 pulses per second), a pattern also observed for other *Allobates* species (*e.g. A. bacurau*, *A. carajas*, *A. crombiei*). However, the advertisement call of the new

species present categorical distinctions when compared to the aforementioned congeners as well as the remaining *Allobates* species occurring throughout Brazil. The advertisement call of the *A. tapajos* population from the study area presented a very similar call structure when compared to the original description (*Lima, Simões & Kaefer, 2015*), except by the more frequent emission of calls composed of three notes and the rarer calls with four notes (more common emission of calls composed of two notes and maximum of three notes according to the original description).

## Genetics

Addressing the internal relationships of the *Allobates* genus was beyond the scope of the present study, and detailed information is provided only for the lineages treated here. The 16S mtDNA gene tree recovered with ML (Fig. S1) and BI (Fig. S2) are similar and generally congruent, but several nodes were weakly supported. In both analyses, the new species was recovered within a more exclusive clade (ML 0.60 of bootstrap; BI 0.90 of posterior probability) including *A. grillicantus* and *A. grillisimilis* + *A. caeruleodactylus* and the new species (Fig. 2). The new species was recovered as sibling of *A. caeruleodactylus* with a low support (ML 0.79 of bootstrap; BI 0.67 of posterior probability). The genetic distance between the new species and *A. caeruleodactylus* was 14.6%, whereas the distance among the new species and the remaining congeners included in our dataset varied between 13.2% and 21.3%. Both species delimitation methods yielded similar results, with the PTP method recovering 70 evolutionary entities (Fig. S1), while the GMYC recovered 79 evolutionary entities (Fig. S2). Both analyses confidently recovered *A. paleci* sp. nov. as an independent evolutionary entity. Sequences of the sympatric population of *A. tapajos* were recovered within a clade with different *A. tapajos* lineages previously reported by *Réjaud et al. (2020)* with a high support (ML 0.98 of bootstrap; BI 1.00 of posterior probability) (Fig. 2). Our *A. tapajos* specimens were recovered as siblings with *A. aff. tapajos* 3 (*sensu Réjaud et al., 2020*) (ML 0.79 of bootstrap; BI 0.96 of posterior probability). The genetic distance between the Teles Pires population and the topotypical sequences was 6.7%. This population was also recovered as an independent evolutionary entity by the PTP and GMYC.

In summary, we found just minor variations regarding morphology and acoustic parameters between the *A. tapajos* originally described and the population from the Teles Pires River. Besides, a notable genetic variation was observed, highlighting the need of a more comprehensive taxonomic evaluation to assert the status of the lineages currently known for *A. tapajos*. The overall morphology, call, and 16S evidences of the remaining *Allobates* from the study area supports it as an undescribed taxon. Hence, we describe it as a new species of *Allobates* from the south Amazonia.

**Taxonomy**
***Allobates paleci* sp. nov.**
Figures 3–7, Tables 1–4.

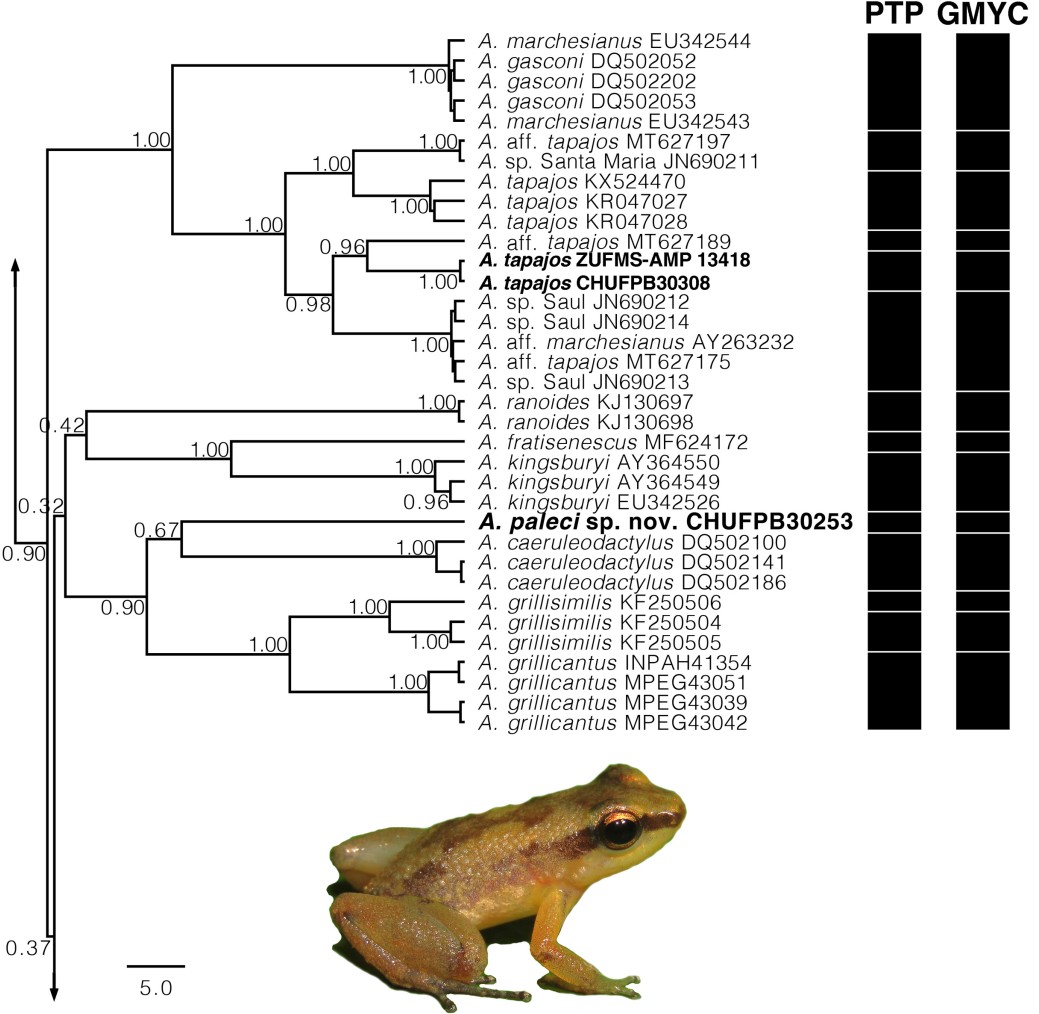

**Figure 2 Gene tree chronogram.** Inset of the Bayesian Inference based on 16S mtDNA gene with the focal *Allobates* lineages of the present study. Section of the phylogenetic analysis of the 16S mtDNA gene for the genus *Allobates*. Nodes are labeled with the Bayesian posterior probability. Black bars represent each evolutionary entities delimited by the following methods: PTP (Poisson Tree Process) and GMYC (Generalized Mixed Yule Coalescent).

**Holotype.** CHUFPB30253. Adult male collected by L. A. Silva and H. Folly, on 17 February 2019, on the right bank of the Teles Pires River, Jacareacanga municipality, Pará state, Brazil (−9.2583°, −56.8057°; datum = WGS84; 184 m above sea level).

**Paratypes.** Eight adult males: CHUFPB30244–45, CHUFPB30248, CHUFPB30251–52, CHUFPB30256, CHUFPB30281, CHUFPB30306; one adult female CHUFPB30242; all collected by L. A. Silva and H. Folly between 16 and 17 February 2019.

**Etymology.** Indigenous populations of the Apiaká ethnic group historically inhabited areas along the major tributaries of the Tapajós, Juruena, Teles Pires, and Arinos rivers. Different familial groups of the Apiaká are called according to their residence regions, and families inhabiting the middle Teles Pires River are known as Paleci. The specific epithet

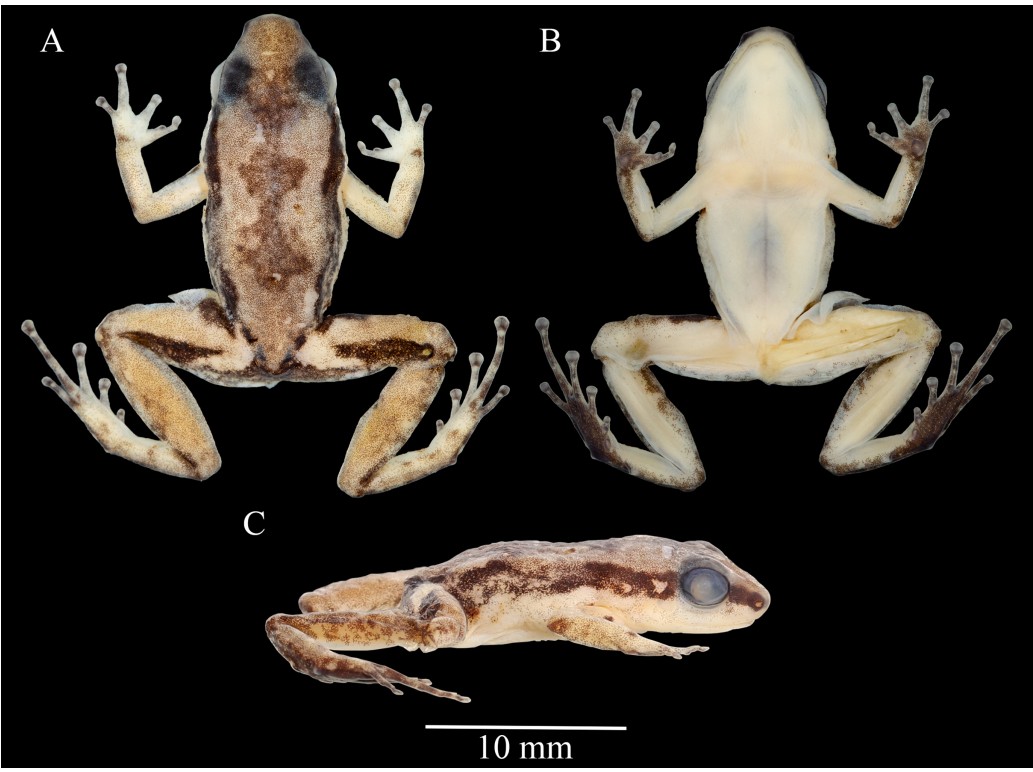

**Figure 3** *Allobates paleci* sp. nov. holotype (CHUFPB30253) in dorsal (A), ventral (B) and lateral (C) views.

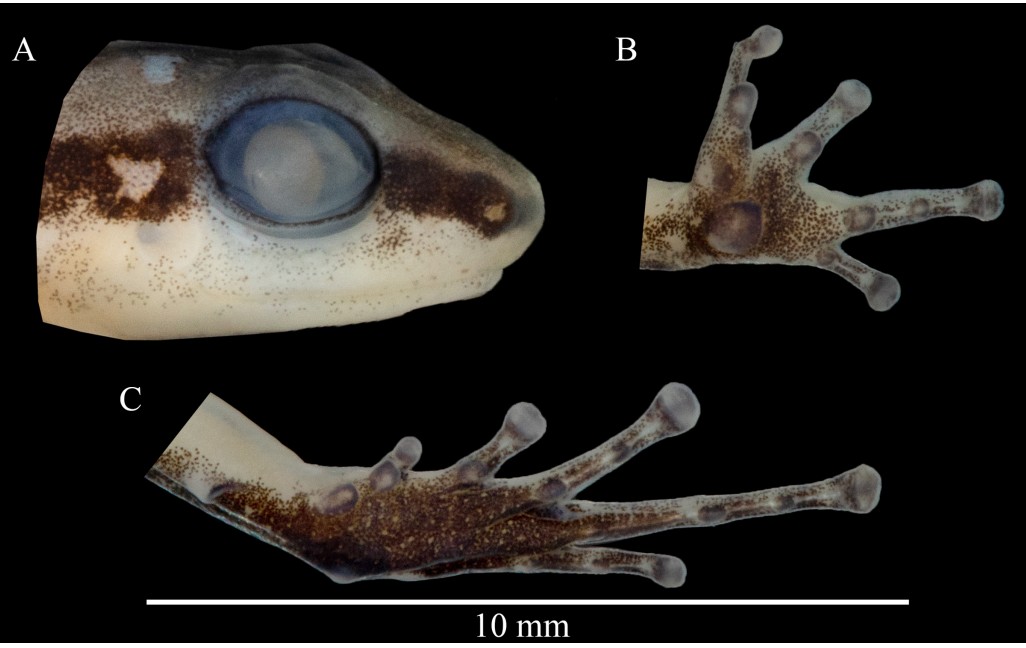

**Figure 4** Head profile (A), ventral left hand (B) and foot (C) of the holotype of *Allobates paleci* sp. nov. (CHUFPB30253).

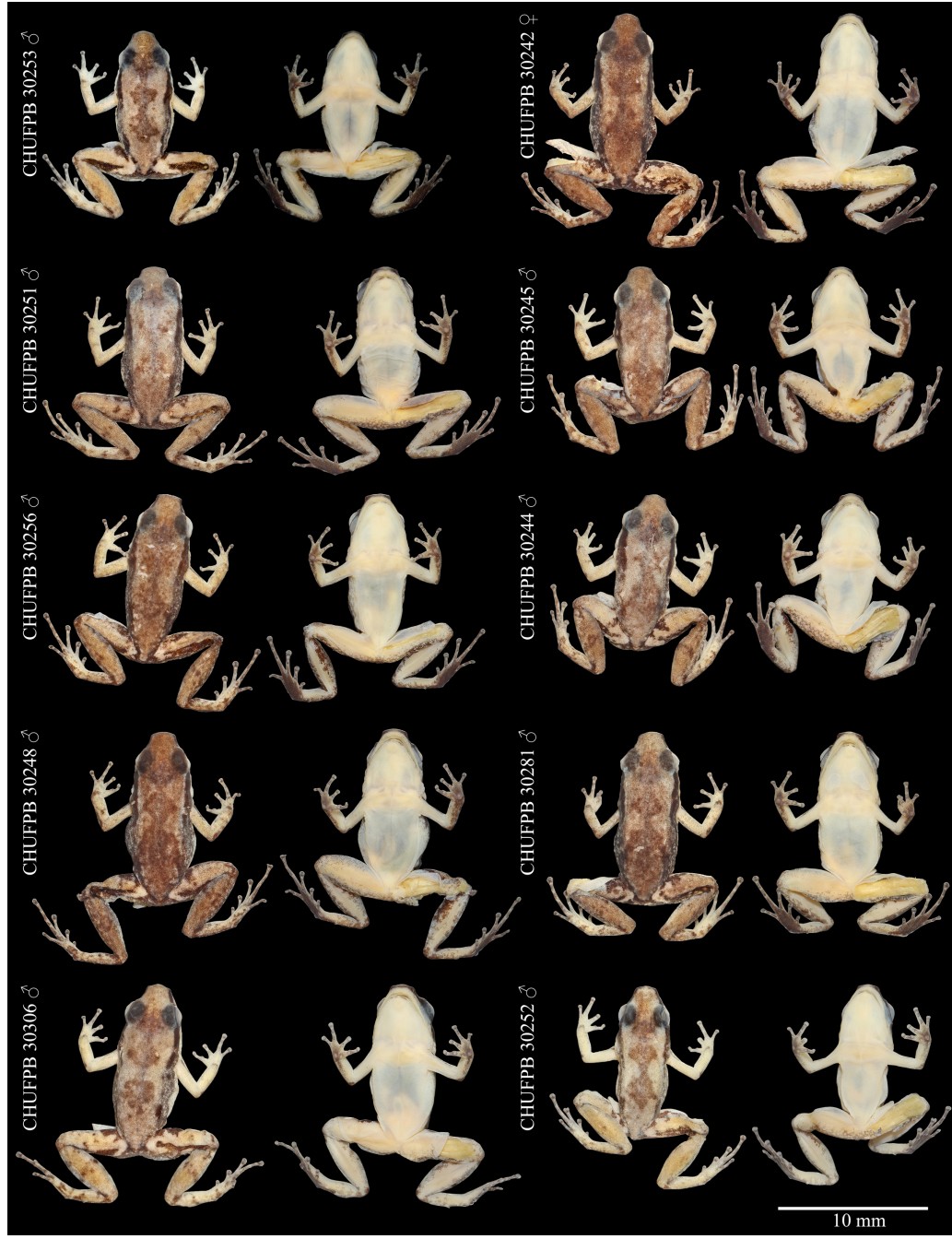

**Figure 5 _Allobates paleci_ sp. nov. type series variation.** Voucher number and sex of each specimen in dorsal (left) and ventral (right) views.

"paleci" is a noun in apposition referring to these families, who live on the vicinities of the new species' type locality. We also suggest the following Portuguese vernacular name for the new species: "sapinho-foguete-dos-paleci".

**Description of the holotype.** Adult male, CHUFPB 30253, SVL = 13.4 mm (Figs. 3 and 4), other measurements are detailed in Table 1. Skin texture slightly granular on dorsum and

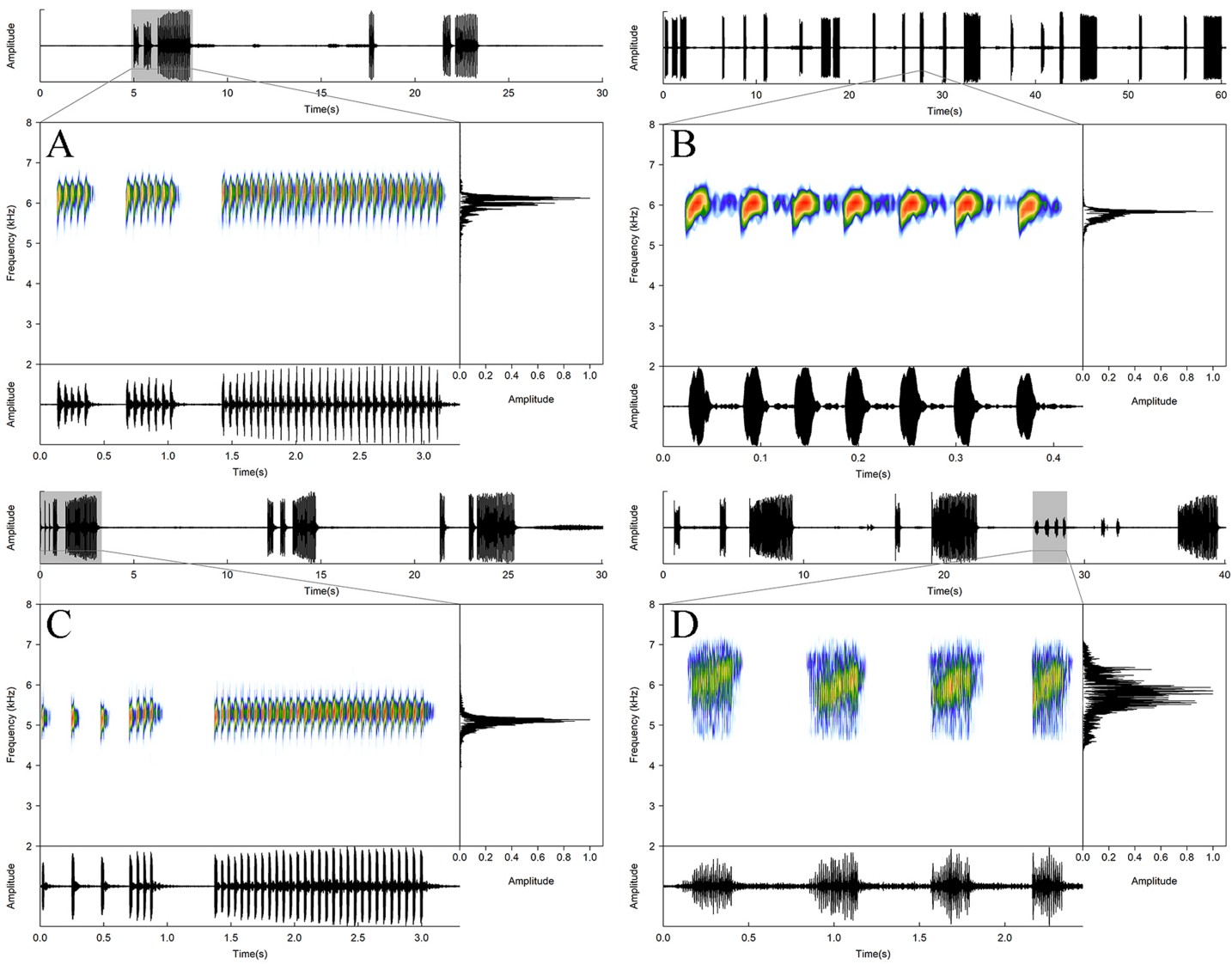

**Figure 6 Acoustic repertory of *Allobates paleci* sp. nov.** Advertisement call of the holotype (CHUFPB30253) (A), warm-up call of an unvouchered specimen (B), solitary notes emitted by the specimen CHUFPB30245 (C), and multipulsed call emitted by the specimen CHUFPB30256 (D).

limbs, smooth on venter. Head wider than longer; head length 77% of head width; head width and head length 40% and 27.7% of SVL, respectively. Interorbital distance 71% of head width. Eye diameter 1.2 times longer than eye-nostril distance; eye diameter 49% of head length. Tympanum round with smooth margins, barely visible to the naked eye. Snout slightly rounded in dorsal view, nearly truncate; snout rounded in lateral view; snout length (eye-nostril distance + nostril–snout distance) 49% of head length. Nostrils located laterally at the tip of the snout; internostril distance 44% of head width. Canthus rostralis from the tip of the snout to the anterior corner of the eye, barely defined. Loreal region vertical. Vocal sac single and subgular. Vomerine teeth absent; maxillary teeth visible under 50× magnification. Choanae located laterally, anterior to eye bulge.

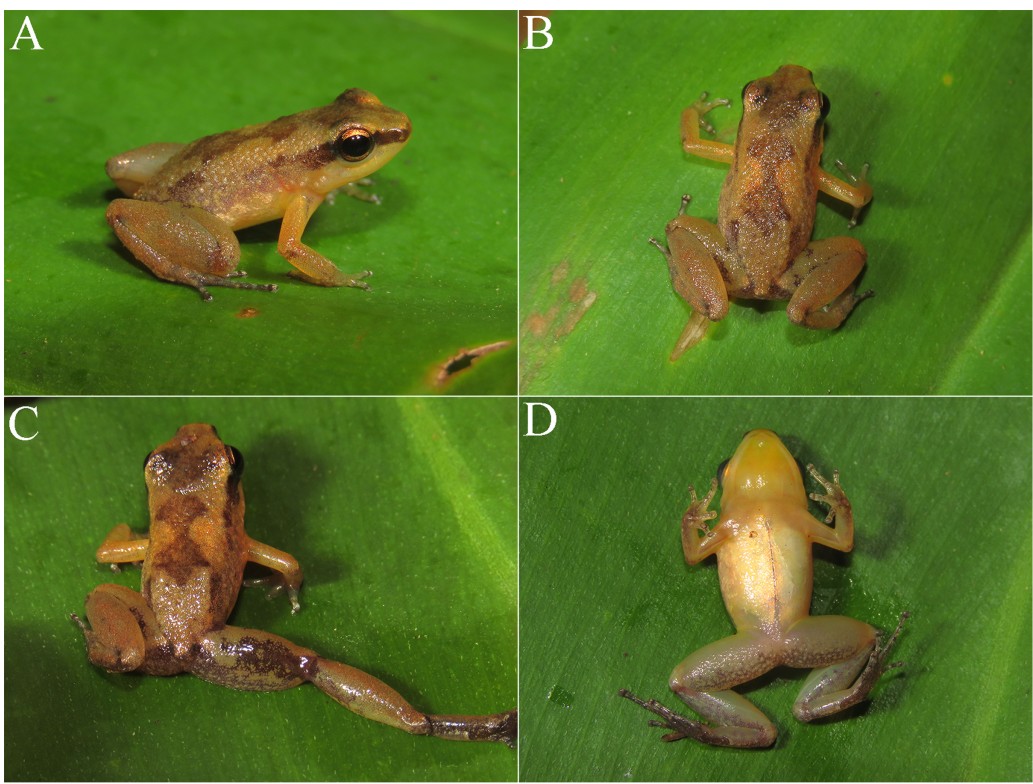

**Figure 7** *Allobates paleci* **sp. nov. in life (CHUFPB 30252), Jacareacanga municipality, Pará state, Brazil.**

Vocal slits conspicuous, laterally located. Tongue longer than wider, attached in the anterior portion of the jaw. Cloacal tubercles absent.

Upper arm length 22% of SVL; forearm length 21% of SVL; upper arm slightly thicker than forearm. Hand without fringes or webbing. Palmar tubercle conspicuous, round to slightly elliptical. Thenar tubercle present, elliptical, less conspicuous than palmar tubercle and half of its size. Subarticular tubercles of fingers III and IV smaller than the width of the finger; subarticular tubercles of fingers II and III round and protuberant. Distal subarticular tubercle presents on finger V, small and round. Supernumerary tubercles and accessory palmar tubercles absent. Metacarpal fold absent. Fingers III and V do not reach the distal subarticular tubercle of finger IV when fingers are adpressed; relative fingers length: IV > II > III > V. Finger IV not swollen. Discs of fingers II–V moderately expanded; width of finger IV disc 75% the size of finger IV third phalanx.

Thigh length and tibia length of similar size, 51% of SVL size each. Tarsal keel present, tubercle-like, softly curved. Inner metatarsal tubercle present, elliptical and conspicuous. Subarticular tubercle of toe I slightly smaller than inner metatarsal tubercle, similar in size to the width of toe I; subarticular tubercles of toes II–IV round and protuberant. Tip of toe I not reaching mid-level of subarticular tubercle of toe II when toes are adpressed; tip of toe III reaching past the proximal subarticular tubercle of toe IV; tip of toe V reaching past one third of the third phalange of toe IV. Metatarsal fold absent. Basal webbing

**Table 1 Morphometric measures of the type series of *Allobates paleci* sp. nov., Jacareacanga municipality, southern Pará state.**

| Measures (mm) | Holotype | Males (*n* = 9) | Female |
| --- | --- | --- | --- |
| Snout vent length | 13.4 | 13.4–16.2 (15.1 ± 1.0) | 16.1 |
| Head length | 3.7 | 3.7–4.8 (4.3 ± 0.3) | 4.5 |
| Head width | 4.8 | 4.7–5.5 (5.2 ± 0.3) | 5.5 |
| Nostril-snout length | 0.4 | 0.3–0.7 (0.5 ± 0.1) | 0.6 |
| Eye-nostril distance | 1.4 | 1.2–1.4 (1.3 ± 0.1) | 1.6 |
| Internarinal distance | 2.1 | 2.0–2.4 (2.2 ± 0.1) | 2.3 |
| Eye diameter | 1.8 | 1.8–2.2 (2.18 ± 0.1) | 2.1 |
| Interorbital distance | 3.4 | 3.4–3.8 (3.6 ± 0.2) | 3.9 |
| Tympanum diameter | 0.7 | 0.6–0.9 (0.8 ± 0.1) | 0.9 |
| Forearm length | 2.8 | 2.8–3.3 (3.1 ± 0.2) | 3.2 |
| Upper arm length | 2.9 | 2.9–3.7 (3.3 ± 0.3) | 3.6 |
| Hand II | 2.0 | 2.0–2.8 (2.5 ± 0.2) | 2.7 |
| Hand III | 2.3 | 2.3–2.7 (2.5 ± 0.1) | 2.5 |
| Hand IV | 3.2 | 3.2–3.8 (3.6 ± 0.2) | 3.5 |
| Hand V | 2.2 | 2.2–2.7 (2.5 ± 0.2) | 2.6 |
| Width of FIV disc | 0.3 | 0.3–0.5 (0.4 ± 0.1) | 0.4 |
| Width of FIV third phalanx | 0.4 | 0.3–0.5 (0.4 ± 0.1) | 0.3 |
| Palmar tubercle diameter | 0.6 | 0.4–0.6 (0.6 ± 0.1) | 0.6 |
| Thenar tubercle diameter | 0.3 | 0.2–0.3 (0.3 ± 0.0) | 0.3 |
| Thigh length | 6.8 | 6.8–8.4 (7.4 ± 0.5) | 7.6 |
| Tibia length | 6.8 | 6.8–7.7 (7.4 ± 0.3) | 6.9 |
| Foot length | 6.0 | 5.9–6.8 (6.3 ± 0.4) | 6.8 |
| Width of toe disc IV | 0.4 | 0.2–0.5 (0.4 ± 0.1) | 0.4 |

present between toes II–III and III–IV. Basal webbing absents between other toes. Relative toe length: IV > III > V > II > I. Discs of toes II–V moderately expanded.

In preservative, dorsum cream, with small dark brown granules from the tip of the snout to the vent region, with a dark brown hourglass mark at the center. Dorsolateral light stripe absent. Dark brown lateral stripe present, strongly pigmented at snout and behind eyes, fading towards the lateral of the body, and becoming pigmented again in the inguinal region. Ventrolateral stripe indistinct. Arms and legs cream, as pigmented as the dorsum background; legs more pigmented than arms. In dorsal view, anterior and posterior region of thigh with dark brown markings. Gular region, chest, belly, upper arms, and thigh cream in ventral view, with only few melanophores in the jaw; forearm, tibia and tarsal region cream with melanophores in ventral view, except at anterior and posterior margins. Palmar and thenar surfaces dark brown.

**Variation.** The single collected female (SVL = 16.2 mm) is larger than 90% of all males of the type series, suggesting the existence of sexual dimorphism in body proportions (Table 1). Complete morphometric variation of the type series is presented in Table 1.

**Table 2 Acoustic parameters of the advertisement (nine recordings, 61 calls, and 549 notes) and warming-up (nine recordings, 72 calls, and 216 notes) calls of *Allobates paleci* sp. nov.** Temporal parameters—call duration (CD), silent interval between the advertisement calls (SIA), number of notes per call (NNC), duration of first three notes (FND), silent interval between the first three notes (FINI), duration of the most central three notes (CND), silent interval between the three most central notes (CINI), duration of last three notes (LND), silent interval between the three last notes (LINI), and note repetition rate (NRR). Spectral parameters—dominant, minimum and maximum frequencies of the whole call (DFC, MinFC, MaxFC), first three notes (DFFN, MinFFN, MaxFFN), most central three notes (DFCN, MinFCN, MaxFCN), and last three notes (DFLN, MinFLN, MaxFLN). The values correspond to range (mean ± standard deviation).

| Call parameters | *Allobates paleci* sp. nov. | |
|---|---|---|
| | Advertisement call | Warming up call |
| CD (s) | 0.97–3.57 (2.29 ± 0.65) | 0.14–0.79 (0.34 ± 0.14) |
| SIA (s) | 8.06–55.83 (16.36 ± 9.88) | 0.21–22.72 (4.72 ± 5.26) |
| NNC | 17–61 (39.93 ± 11.18) | 3–15 (6.65 ± 2.35) |
| DFC (Hz) | 5,167.97–6,201.56 (5,717.24 ± 220.81) | 4,995.7–6,029.30 (5,663.23 ± 243.51) |
| MinFC | 4,995.70–5,943.16 (5,463.08 ± 218.73) | 4,823.44–5,857.03 (5,438.33 ± 223.63) |
| MaxFC | 5,340.23–6,287.70 (5,886.68 ± 218.4) | 5,167.97–6,201.56 (5,854.64 ± 235.76) |
| FND (ms) | 13.00–41.10 (27.79 ± 6.86) | 23.40–48.50 (33.15 ± 5.98) |
| CND (ms) | 14.50–48.20 (29.08 ± 7.58) | 15.20–55.10 (30.18 ± 7.2) |
| LND (ms) | 15.20–50.20 (30.23 ± 8.56) | 17.60–51.20 (30.48 ± 9.08) |
| NRR | 16.32–19.10 (17.49 ± 0.68) | 17.34–23 (20.06 ± 1.29) |
| DFFN (Hz) | 4,995.70–6,201.56 (5,668.76 ± 239.37) | 4,995.70–6,029.30 (5,640.50 ± 256.57) |
| MinFFN | 4,823.44–5,857.03 (5,440.02 ± 228.64) | 3,273.05–5,857.03 (5,380.91 ± 345.13) |
| MaxFFN | 5,167.97–6,373.83 (5,845.74 ± 231.69) | 5,167.97–6,201.56 (5,833.11 ± 249.23) |
| DFCN (Hz) | 5,167.97–6,201.56 (5,714.89 ± 215.34) | 4,995.70–6,201.56 (5,669.21 ± 240.74) |
| MinFCN | 4,995.70–5,857.03 (5,470.14 ± 211.34) | 4,995.70–5,857.03 (5,471.83 ± 220.65) |
| MaxFCN | 5,340.23–6,373.83 (5,888.10 ± 217.09) | 5,340.23–6,373.83 (5,853.44 ± 232.4) |
| DFLN (Hz) | 4,995.70–6,201.56 (5,699.83 ± 222.86) | 4,995.70–6,029.3 (5,638.11 ± 235.75) |
| MinFLN | 4,995.70–6,029.30 (5,461.67 ± 219.71) | 4,995.70–5,857.03 (5,443.12 ± 217.15) |
| MaxFLN | 5,340.23–6,373.83 (5,873.98 ± 216.79) | 5,167.97–6,201.56 (5,811.57 ± 233.09) |
| FINI (ms) | 12.6–37.4 (24.28 ± 6.01) | 10.70–33.50 (22.84 ± 5.08) |
| CINI (ms) | 14.6–48.4 (29.02 ± 7.77) | 11.4–36.6 (25.05 ± 5.3) |
| LINI (ms) | 15.1–56.1 (33.35 ± 8.41) | 21.1–44.7 (32.79 ± 4.96) |

Variation in coloration is more evident in preserved specimens (Fig. 5). The concentration of small dark brown granules varies, as only the holotype and three other males showed cream coloration on dorsum, while the remaining specimens show more concentration of these granules, having a brownish coloration; the female have the dark dorsum typical of other specimens in the type series. Beside the holotype, five other males have a well-defined pigmented hourglass-shaped mark at the center of dorsum. In contrast, other three males and the female have this dark hourglass mark indistinct, as their dorsum background are also darker. Three specimens have thighs cream with sparse melanophores in ventral view (immaculate cream in the remaining specimens). Tibia coloration in ventral view is cream at the center and the margins range from almost no

**Table 3 Single notes (one recording, three calls—voucher CHUFPB30245) and multipulsed singular notes (one recording, six calls—voucher CHUFPB30256) emitted by *Allobates paleci* sp. nov.: call duration (CD), silent interval between calls (SIC), number of pulses per call (NPC), and dominant, minimum and maximum frequencies of the call (DF, MinDF, MaxDF).** The values correspond to range (mean ± standard deviation).

| Call parameters | *Allobates paleci* sp. nov. | |
| --- | --- | --- |
| | **Single note** | **Single multipulsed notes** |
| CD (ms) | 15.30–26.10 (21.97 ± 5.83) | 180–340 (240 ± 6) |
| SIC (ms) | 196.90–215.30 (206.10 ± 9.20) | 360–2,560 (930 ± 930) |
| NPC | – | 16–33 (22 ± 6.26) |
| DF (Hz) | 4,995.70–4,995.70 (4,995.70 ± 0) | 5,512.50–6,029.30 (5,828.32 ± 169.37) |
| MinDF | 4,823.44–4,823.44 (4,823 ± 0) | 5,167.97–5,512.50 (5,340.23 ± 108.95) |
| MaxDF | 5,168.97–5,168.97 (5,168 ± 0) | 6,201.56–6,373.83 (6,230.27 ± 70.33) |

**Table 4 Acoustic parameters of the advertisement call of *Allobates tapajos* population from the Teles Pires River (nine recordings, 186 calls and 315 notes): call duration (CD), silent interval between the advertisement calls (SIA), number of notes per call (NNC), note repetition rate (NRR), note duration (ND), silent interval between notes (SIN).** Spectral parameters – dominant, minimum and maximum frequencies of the note (DFC, MinFN, MaxFN). The values correspond to range (mean ± standard deviation).

| Call parameters | Range (Mean ± Standard deviation) |
| --- | --- |
| CD (ms) | 60–460 (260 ± 70) |
| SIA (ms) | 290–740 (440 ± 70) |
| NNC | 1–4 (2.6 ± 0.59) |
| DFC (Hz) | 5,512.5–6,029.3 (5,757.48 ± 119.55) |
| NRR/s | 6.58–16.39 (10.03 ± 1.24) |
| ND (ms) | 30–60 (40 ± 0) |
| SIN (ms) | 60–730 (100 ± 70) |
| DFN (Hz) | 5,340.2–6,029.3 (5,737.28 ± 141.78) |
| MinFN (Hz) | 4,659.7–5,540.4 (5,031.93 ± 189.73) |
| MaxFN (Hz) | 5,907.8–6,840.1 (6,308.36 ± 256.74) |

pigmentation to highly pigmented with melanophores. The dark brown lateral stripe is well defined at snout and from behind eyes until the arm line, then fades in different degrees toward the inguinal region in each specimen.

Arms are less pigmented in dorsal view in all specimens, with irregular blotches on the forearm near the elbow and near the hand. Hands in dorsal view are as pigmented as the arms. All specimens presented dark brown patches at the anterior and posterior region of thigh. Patches in thigh are of irregular shape, either restricted as a stripe on the anterior portion of the thigh or extending towards the posterior portion of the thigh. Small blotches of dark coloration are observed in tibia, tarsus, and foot in dorsal view.

The only specimen from the type series photographed in life was the male CHUFPB 30252 (Fig. 7). Nevertheless, the remaining type series was generally concordant with the

coloration pattern observed for this individual. In life, the specimen CHUFPB 30252 (SVL = 14.3 mm) had the dorsum of the body light brown, with a dark brown hourglass mark at the center. Dorsolateral stripe absent. Dark brown lateral stripe present, strongly pigmented at snout and behind eyes, fading towards the inguinal region. Light golden ventrolateral stripe interrupted, more evident at the medial portion of the body towards the inguinal region. Arms and legs light brown as pigmented on the dorsum. Anterior and posterior region of thigh with longitudinal dark brown patches. Gular region yellow without obvious melanophores, chest yellowish, belly yellowish with a white subjacent peritoneum. Upper arms in ventral view yellowish. Thigh in ventral view yellowish at distal portion becoming whitish towards the insertion of legs; few iridophores along the lower margin of the thigh's ventral surfaces; forearm, tibia and tarsal regions light yellowish in ventral view with melanophores, except at anterior and posterior margins. Ventral surface of foot dark brown; ventral surface of hand with a dark brown pigmentation on the palmar region and around tubercles, fingers light brown scattered with few melanophores.

**Generic placement.** The new species is assigned to the genus *Allobates* based on the presence of the following morphological characteristics: Finger V length not reaching the distal subarticular tubercle of Finger IV, basal webbing with lateral fringe on the preaxial side of Toe IV, presence of pale paracloacal marks, presence of a pale ventrolateral stripe, and the presence of a diffuse oblique lateral stripe (*Grant et al., 2017*). Besides, the molecular data (Figs. S1 and S2) placed the new species as sister taxa of other *Allobates* from Amazonia.

**Diagnosis.** *Allobates paleci* **sp. nov.** can be distinguished from the other species of the genus by the following combination of characters: (1) small size (males SVL 13.4–16.2 mm (15.1 ± 1.0), female SVL 16.07 mm); (2) dorsal surface of thighs light brown lacking dark brown transversal bars, abdomen immaculate yellowish in life; (3) dorsum light brown with dark brown hourglass mark ranging from the interorbital level to the urostyle region; (4) gular region of males yellowish in life, lacking obvious melanophores; (5) interrupted dark lateral stripe, more conspicuous anteriorly; (6) advertisement calls formed by trills with a duration of 0.97–3.57 s (2.29 ± 0.65), 17–61 notes (39.93 ± 11.18) emitted at a rate of 16.32–19.10 notes per second (17.49 ± 0.68), and dominant frequency ranging between 5,168 and 6,202 Hz (5,717 ± 220.81).

**Morphological comparisons.** The new species described here is only known to occur in ombrophilous forests from southern Brazilian Amazonia, in the boundary of the southern Pará state with northern Mato Grosso state, Brazil. Based on this restricted distribution and molecular phylogenetic affinities, we phenotypically compare the new species with all 31 Brazilian congeners (54.4% of the currently known genus diversity): *Allobates bacurau Simões, 2016*, *A. brunneus*, *A. caeruleodactylus* (*Lima & Caldwell, 2001*), *A. caldwellae Lima, Ferrão & Silva, 2020*, *A. carajas*, *A. conspicuus* (*Morales, 2002*), *A. crombiei*, *A. femoralis*, *A. flaviventris Melo-Sampaio, Souza & Peloso, 2013*, *A. fuscellus* (*Morales, 2002*), *A. gasconi* (*Morales, 2002*), *A. goianus* (*Bokermann, 1975*), *A. grillicantus* (*Moraes & Lima, 2021*), *A. grillisimilis*, *A. hodli*, *A. magnussoni Lima, Simões & Kaefer,*

*2014*, *A. marchesianus* (*Melin, 1941*), *A. masniger* (*Morales, 2002*), *A. myersi*, *A. nidicola* (*Caldwell & Lima, 2003*), *A. nunciatus Moraes, Pavan & Lima, 2019*, *A. olfersioides*, *A. pacaas Melo-Sampaio et al., 2020*, *A. paleovarzensis Lima et al., 2010*, *A. subfolionidificans* (*Lima, Sanchez & Souza, 2007*), *A. sumtuosus* (*Morales, 2002*), *A. tapajos*, *A. tinae Melo-Sampaio, Oliveira & Prates, 2018*, *A. trilineatus* (*Boulenger, 1884*), *A. vanzolinius* (*Morales, 2002*), and *A. velocicantus Souza et al., 2020*. The character states of the compared species are given in parentheses. *Allobates paleci* **sp. nov.** can be easily distinguished from *A. femoralis*, *A. hodli*, and *A. myersi* by having dorsal surface of the thigh light brown lacking dark brown transversal bars, abdomen immaculate and yellowish in life (red or yellow flash mark on dorsal surface of thigh, and black and white marbling on the abdomen) (*Boulenger, 1884*; *Pyburn, 1981*; *Simões, Lima & Farias, 2010*).

From *A. bacurau*, *A. caeruleodactylus*, *A. caldwellae*, *A. conspicuus*, *A. fuscellus*, *A. grillicantus*, *A. grillisimilis*, *A. juami*, *A. masniger*, *A. nidicola*, *A. nunciatus*, *A. paleovarzensis*, *A. subfolionidificans*, *A. sumtuosus*, *A. tinae*, *A. trilineatus*, *A. vanzolinius*, and *A. velocicantus*, *A. paleci* **sp. nov.** can be distinguished by have a light brown dorsum with dark brown hourglass mark ranging from the interorbital level to the urostyle region (dorsum without contrasting marks) (*Melin, 1941*; *Lima & Caldwell, 2001*; *Morales, 2002*; *Caldwell & Lima, 2003*; *Lima, Sanchez & Souza, 2007*; *Lima et al., 2010*; *Simões et al., 2013a*; *Simões et al., 2013b*; *Simões, 2016*; *Melo-Sampaio, Oliveira & Prates, 2018*; *Simões et al., 2018*; *Moraes, Pavan & Lima, 2019*; *Lima, Ferrão & Silva, 2020*; *Souza et al., 2020*). *Allobates paleci* **sp. nov.** is distinguished from *A. flaviventris*, *A. gasconi*, *A. magnussoni*, *A. marchesianus*, and *A. pacaas* by presenting yellowish throat in live males, free of obvious melanophores (throat light gray to dark grey) (*Boulenger, 1884*; *Morales, 2002*; *Melo-Sampaio, Souza & Peloso, 2013*; *Lima, Simões & Kaefer, 2014*; *Melo-Sampaio et al., 2020*). *Allobates paleci* **sp. nov.** presents dorsal surface of thighs lacking dark brown transversal bars and thus is distinguished from *A. brunneus*, *A. carajas*, *A. crombiei*, *A. goianus*, and *A. olfersioides* (dark brown transverse bands/blotches on thigh) (*Cope, 1887*; *Lutz, 1925*; *Bokermann, 1975*; *Morales, 2002*; *Lima, Caldwell & Strüssmann, 2009*; *Simões, Rojas & Lima, 2019*).

*Allobates tapajos* is the most morphologically/chromatically similar species, but the new species presents an interrupted dark lateral stripe, more conspicuous anteriorly (dark lateral stripe gradually fades towards the inguinal region) (*Lima, Simões & Kaefer, 2015*). Furthermore, *A. paleci* **sp. nov.** and nominal *A. tapajos* are acoustically and molecularly very distinct (see below).

**Call description.** Advertisement calls (*n* = 61, nine males) of *Allobates paleci* **sp. nov.** (Fig. 6A, Table 2) are characterized by relatively long trills of 17–61 notes (39.9 ± 11.9) emitted at a rate of 16.3–19.1 pulses per second (17.49 ± 0.68), and call duration ranging between 0.97 and 3.57 s (2.29 ± 0.649). The call is irregularly emitted between intervals lasting 8.06–55.83 s (16.36 ± 9.88). The durations of the first, central and last notes are similar: 13.00–41.10 ms (27.79 ± 6.86), 14.50–48.20 ms (29.08 ± 7.58), and 15.20–50.20 ms (30.23 ± 8.56), respectively. The intervals of pulses from begin, middle and final portion of the call are also similar: 12.60–37.40 ms (24.28 ± 6.01), 14.60–48.40 ms (29.02 ± 7.77),

and 15.10–56.10 ms (33.35 ± 8.41), respectively. Furthermore, each note was composed of two or three visible pulses. The amplitude modulation along the call is homogeneous. The dominant frequency of the call range between 5,168 and 6,202 Hz (5,717 ± 221). As observed for the call amplitude modulation, the frequency is also maintained through the entire call, with the first, central and last notes similar in dominant frequencies: 4,996–6,202 Hz (5,669 ± 239), 5,168–6,202 Hz (5,715 ± 215), and 4,996–6,202 Hz (5,700 ± 223), respectively.

Warming-up calls (*n* = 72, nine males) of *Allobates paleci* **sp. nov.** (Fig. 6B and Table 2) are composed of short trills of 3–15 notes (6.65 ± 2.35) emitted at a rate of 17.35–23.00 pulses per second (20.06 ± 1.29), and call duration ranging between 0.14 and 0.79 s (0.34 ± 0.14). The call is irregularly emitted between intervals lasting 0.21–22.73 s (4.72 ± 5.26). The durations of the first, central and last notes were similar, 23.40–48.50 ms (33.15 ± 5.98), 15.20–55.10 ms (30.18 ± 7.20), and 17.60–51.20 ms (30.48 ± 9.08), respectively. The interval of pulses from begin, middle and final portion of the call were also similar, 10.70–33.50 ms (22.84 ± 5.08), 11.40–36.60 ms (25.05 ± 5.30), and 21.10–44.70 ms (32.79 ± 4.96), respectively. Each note is composed of two or three pulses. The amplitude modulation along the call is homogeneous. The dominant frequency of warming-up calls range between 4,996 and 6,029 Hz (5,663 ± 243.51). As observed for the call amplitude modulation, the frequency was also maintained through the entire call, with the first, central and last notes having similar dominant frequencies: 4,996–6,029 Hz (5,641 ± 256.57), 4,996–6,202 Hz (5,669 ± 240.74), and 4,996–6,029 (5,638 ± 235.75), respectively.

Additionally, we recorded the male CHUFPB30245 (field number AAGARDA12596) emitting three consecutive calls composed of a single unpulsed note before the emission of a warming-up call (Fig. 6C and Table 3). These calls last for 15.30–26.10 ms (21.97 ± 5.83), are emitted at a rate of 196.90–215.30 calls per ms (206.10 ± 9.20), and have a dominant frequency of 4,996 Hz. We also recorded the male CHUFPB30256 (field number AAGARDA12595) emitting six calls composed of single multipulsed notes at the end of an advertisement call emission series (Fig. 6D and Table 3). These calls last for 0.18–0.34 s (0.24 ± 0.06), are emitted at a rate of 0.36–2.56 calls per second (0.93 ± 0.93), have between 16 and 33 pulses (22.00 ± 6.26), and a dominant frequency of 5,513–6,029 Hz (5,828 ± 169.37). We observed two other males (both unvouchered) emitting this last call type just before moving through the leaflitter (one of them can be seen in the following footage: https://youtu.be/WvngJ1tEMYI). Although no other individual of *Allobates paleci* **sp. nov.** was observed around the calling males emitting these multipulsed notes, we suggest that this call type was emitted in a courtship or territorial context. The acoustic envelope of this call type resembles both the courtship call described for *A. hodli* (*Simões, Lima & Farias, 2010*) and the aggressive call described for of *A. olfersioides* (*Forti, Silva & Toledo, 2017*). Further field observations are needed to clarify the social context of this call type.

**Bioacoustic comparison.** *Allobates paleci* **sp. nov.** has a unique combination of the following acoustic parameters: advertisement calls formed by trills lasting 0.97–3.57 s

(2.27 ± 0.65) with 17–61 notes (39.93 ± 11.18) emitted at 16.32–19.10 notes/s (17.49 ± 0.68), and dominant frequency ranging between 5,168 and 6,202 Hz (5,717 ± 221).This call structure differs from the calls of six compared species, which consist of single notes emitted continuously: *A. caeruleodactylus*, *A. magnussoni*, *A. masniger*, *A. nidicola*, *A. olfersioides*, and *A. subfolionidificans* (*Lima & Caldwell, 2001*; *Caldwell & Lima, 2003*; *Lima, Sanchez & Souza, 2007*; *Tsuji-Nishikido et al., 2012*; *Lima, Simões & Kaefer, 2014*; *Simões, 2016*; *Forti, Silva & Toledo, 2017*). Three other species, in addition to a call type consisting of single notes emitted continuously, may also produce trills of notes, although smaller than in *A. paleci* **sp. nov.**: *A. marchesianus*: trills of 21–24 notes and duration of 3.39–4.40 s (*Caldwell, Lima & Keller, 2002*); *A. sumtuosus*: trills of 23–35 notes and duration of 3.949–5.878 s (*Simões et al., 2013b*); and *A. brunneus*: trills of 6–11 notes and duration of 1.68–4.18 s (*Lima, Caldwell & Strüssmann, 2009*); furthermore, *A. paleci* **sp. nov.** do not emit advertisement calls arranged in single notes. Two species, *A. carajas* and *A. tapajos*, have at least four different temporal call arrangements, also including a trilled call type: *A. carajas* – continuous emission of notes separated by regular silent intervals, continuous emission of notes separated by irregular silent intervals, emission of discrete note trills, and sporadic emission of single notes (*Simões, Rojas & Lima, 2019*); when emitting trills, they reach longer durations (up to 7.05 s) but with a lower note number (up to 22 notes) when compared to *A. paleci* **sp. nov.**; *A. tapajos* possess note pairs (most common arrangement), single notes emitted between note pairs, and note trios (rarest arrangement); just one male emitted trills similar in duration to the calls of *A. paleci* **sp. nov.** (2.46–3.37 s), but with fewer notes per trill (10–14) (*Lima, Simões & Kaefer, 2015*).

The advertisement call of four other species are characterized by short bouts of notes regularly emitted, whereas *A. paleci* **sp. nov.** always have trills with 17–61 notes (39.93 ± 11.18) emitted at 16.32–19.10 notes/s: *A. femoralis* has calls composed of groups of one, three or four notes lasting between 0.044 and 0.053 s (*Amézquita et al., 2009*; *Simões, Lima & Farias, 2010*; *Moraes, Pavan & Lima, 2019*); *A. hodli* has short trills ranging 0.140–0.198 s with two whistle-like notes (*Simões, Lima & Farias, 2010*); *A. myersi* has three different temporal arrangements: short trills (mean 0.35 ± 0.02 s) of two, three, or four notes (*Simões & Lima, 2011*); *A. nunciatus* has trills of up to 0.357 s and of four notes (*Moraes, Pavan & Lima, 2019*).

The 11 remaining species emit only bouts of notes here defined as a trilled advertisement calls, a general temporal pattern of note emission shared with *Allobates paleci* **sp. nov.** Five of these species emit shorter trills and fewer notes per call when compared to *A. paleci* (call duration 0.97–3.57 s and 17–61 notes): *A. caldwellae* has calls ranging between 0.259–1.255 s and 3–7 notes (*Lima, Ferrão & Silva, 2020*); *A. flaviventris* has 2–10 notes within each trill and presumably a shorter call duration (not informed in the original description, *Melo-Sampaio, Souza & Peloso, 2013*); *A. grillicantus* has trills lasting 0.151–0.507 s with 3–15 notes (*Moraes & Lima, 2021*); *A. grillisimilis* has trills ranging 0.122–0.305 s with 3–15 notes (*Simões et al., 2013a*; *Simões, 2016*); and *A. trilineatus* has trills of 0.97–1.55 s with 9–13 notes (*Grant & Rodríguez, 2001*).

Four species presented trills with similar durations and number of notes, but with smaller note repetition rates (note repetition rate of 16.32–19.10, mean of 17.49 ± 0.68 in *Allobates paleci* **sp. nov.**): *A. goianus* calls have trills with a mean duration of 3.9 s and 2–41 notes emitted at 3.1–3.9 notes per second (*Carvalho, Martins & Giaretta, 2016*); *A. paleovarzensis* trills of 0.72–3.02 s with 3–21 notes emitted at 6.97 notes/s (*Lima et al., 2010*); *A. tinae* trills last 0.285–2.27 s (1.50 ± 0.45) with 2–9 notes (mode = 8) emitted at 5.34 notes/s (*Melo-Sampaio, Oliveira & Prates, 2018*). Finally, *A. velocicantus* presented a similar trill duration of 1.87–2.89 s (2.49 ± 0.22), with 66–138 notes emitted at 51.2 ± 5.8 (38.4–56.8) notes/s (*Souza et al., 2020*). Two species presented longer trills with more notes: *A. bacurau* has trills of 7–11 s with 60–81 notes (*Simões, 2016*); and *A. juami* has trills of 2.5–5.09 s (4.51 ± 0.37) with 60–73 notes (65 ± 4) (*Simões et al., 2018*).

*Allobates crombiei* shows the most similar trill regarding the calls emitted by *A. paleci* **sp. nov.** (*Lima, Erdtmann & Amézquita, 2012*). *Allobates crombiei* presents trills with mean duration of 3.52 ± 0.49 s (1.91–4.53) with 43 ± 6.38 notes (25–59), and, being the main distinction compared to the advertisement call of *A. paleci* **sp. nov.**, showed a lower note repetition rate of about 12.21 notes per second (call duration of 0.970–3.574 s (2.286 ± 0.649), 17–61 notes (39.934 ± 11.183), and note repetition rate of 16.32–19.10 (17.49 ± 0.68) in *A. paleci* **sp. nov.**). In respect to spectral parameters, *A. crombiei* has an ascendant but quick frequency modulation, while *A. paleci* **sp. nov.** lacks frequency modulation.

Five of the 31 species have no information regarding their advertisement calls: *A. conspicuus*, *A. fuscellus*, *A. gasconi*, *A. pacaas*, and *A. vanzolinius* (*Morales, 2002*; *Melo-Sampaio et al., 2020*). Nevertheless, the new species is easily distinguished from them based on morphological evidence (see above).

**Geographic distribution and natural history.** *Allobates paleci* **sp. nov.** is known only from the type locality, in the right bank of the Teles Pires River, Jacareacanga municipality, southern Pará state, Brazil. In this locality, the new species was only recorded inside dense ombrophilous forests. During the rainy season (specially between November and February), several males of the new species were found calling from the moist leaf litter. The calling behavior was mainly concentrated in rainy days between 06:00–10:00 am and 04:00–06:00 pm. Despite being recorded in very close sites and after a considerable sampling effort, *Allobates paleci* **sp. nov.** and *A. tapajos* were never found calling syntopically. Furthermore, *A. paleci* **sp. nov.** was less abundant at the type locality when compared to the sympatric *A. tapajos*. More details on reproductive behavior, oviposition, and tadpoles of *Allobates paleci* **sp. nov.** remain unknown.

**Molecular.** Average sequence divergence between the new species and the compared nominal congeners was 16.5%, ranging from 13.2% (*A. carajas*) to 21.3% (*A. niputidea*) (Table S1).

## Remarks on the sympatric *Allobates*

*Allobates tapajos* was described based on individuals collected at Parque Nacional da Amazônia, Itaituba Municipality, Pará state, Brazil (*Lima, Simões & Kaefer, 2015*).

This species is known from areas adjacent to the type locality on the right and left banks of the middle and lower Tapajós River, in different sites of the municipalities of Aveiro, Belterra, and Itaituba, all in Pará state (*Lima, Simões & Kaefer, 2015*; *Maia, Lima & Kaefer, 2017*). The coloration pattern and measurements (mean SVL 14.4 ± 1.7, range 12.2–16.8 mm; *n* = 15 males) of the *Allobates* found in sympatry with *A. paleci* fitted in the variation described in the original description of *A. tapajos* (*Lima, Simões & Kaefer, 2015*) (Fig. 8). Acoustic parameters of this population also overlapped with those described for the nominal *A. tapajos*, except for number of notes, which ranged from 1 to 4, but most commonly presents three notes (calls ranging from 1 to 3 notes for *A. tapajos*, most common pattern described includes calls with two notes, *Lima, Simões & Kaefer, 2015*) (Fig. 8, Table 4). A footage of this species calling can be seen in https://www.youtube.com/watch?v=0fu2JJLRhHo&ab. Based on a mitogenomic approach and a comprehensive geographic sampling, *Réjaud et al. (2020)* reported three undescribed lineages related to *A. tapajos*. In our phylogenetic inferences (BI and ML), the Teles Pires River population was recovered as sibling of *A.* aff. *tapajos* 3 (*sensu Réjaud et al., 2020*), with a sequence divergence of 4.8% between them (Fig. 2, Table S1). When compared to *A. tapajos* from the type locality, the sequence divergence is up to 6.7%. In fact, both GMYC and PTP species delimitation methods recovered the Teles Pires population as an independent lineage. The population here recorded is nearly 560 km in straight line from the type locality of *A. tapajos*.

## DISCUSSION

The use of bioacoustics, morphological, and molecular evidence has proven useful to help identify, diagnose, and describe morphologically similar species of *Allobates* (*e.g. Simões et al., 2018*; *Lima, Ferrão & Silva, 2020*; *Jaramillo et al., 2021*). In fact, our molecular evidence shows that both lineages from the study area were not included in any previous phylogenetic/taxonomic appraisals of the genus. Additionally, the species delimitations methods employed here, using the sequences currently available in the GenBank, indicated an underestimation level of nearly 20.6% (PTP) and 36.2% (GMYC) of *Allobates* species. This scenario reinforces the likelihood of tremendous underestimated diversity of this genus in Amazonia and highlights the importance of the integrative taxonomy to address this problem.

We further stress the need of an objective standardization for advertisement call description for *Allobates*, especially for trilled calls with high rates of note repetition. For example, based on the oscillograms, the same acoustic structure periodically emitted (repetition unity) by cryptic species is considered either as notes (*A. grillisimilis*, *Simões et al., 2013a*) or pulses (*A. grillicantus*, *Moraes & Lima, 2021*). The current lack of an objective framework for advertisement call descriptions hampers comparisons and may even cause noise in the taxonomy of the genus. Because *A. paleci* shows an advertisement call with complete amplitude modulation, including a clear but short silent interval between consecutive repetition units, we refer to them as notes (*Köhler et al., 2017*). Finally, recent studies have proposed such standardization for other anuran groups, such as the review of acoustic traits for *Physalaemus* (*Hepp & Pombal, 2020*), which may represent a useful example of what could be proposed for *Allobates*.

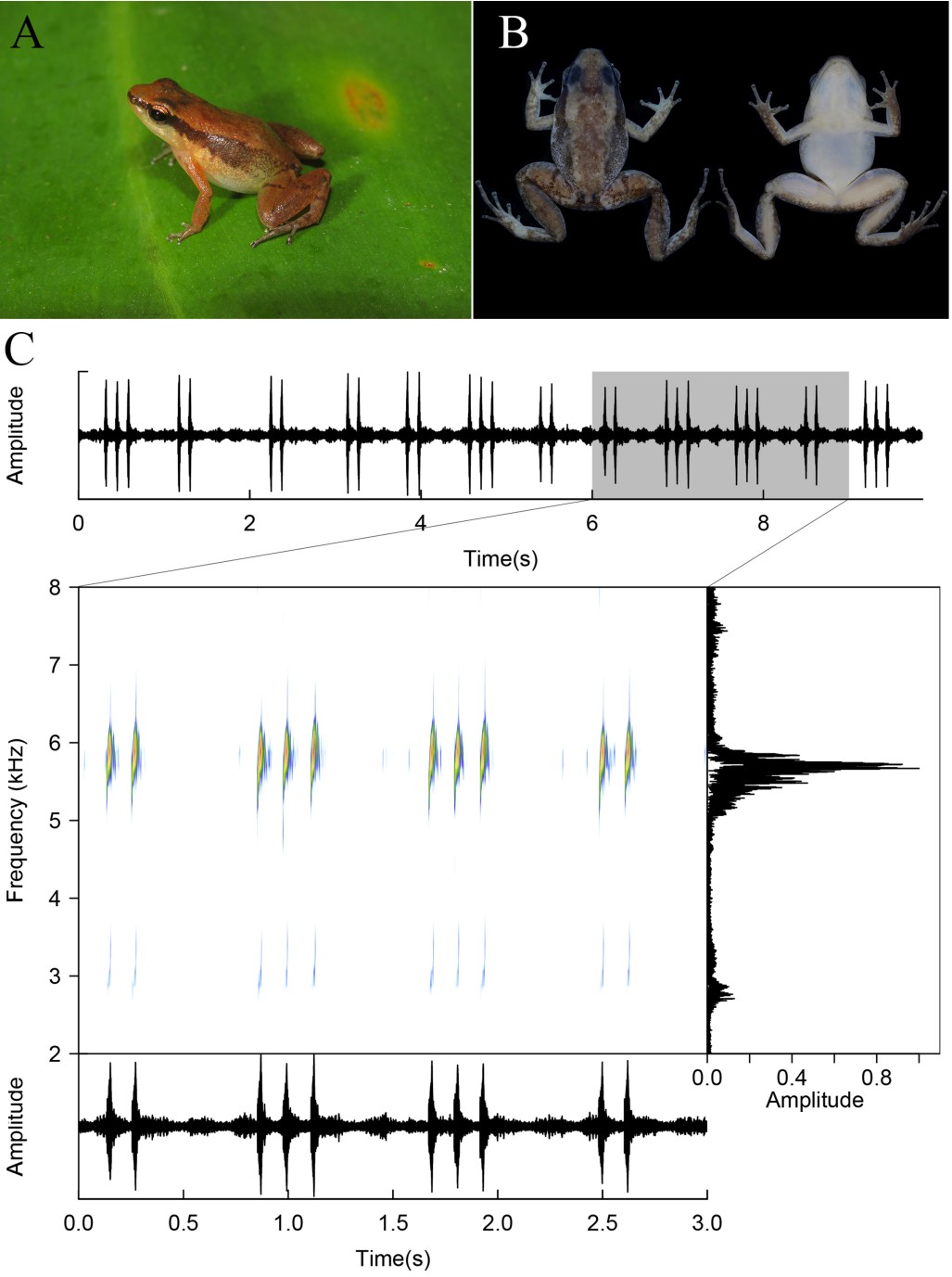

Figure 8 *Allobates tapajos* from the Teles Pires River, Paranaíta, Mato Grosso, Brazil. Specimen in life (A; unvouchered), in preservative (B; ZUFMS7720), and advertisement call (unvouchered).

Both PTP and GMYC species delimitation approaches recovered the *Allobates tapajos* from the Teles Pires River as an independent lineage, which presented 6.2–6.7% of genetic distance from *A. tapajos* from the type locality (Fig. 2, Table S1). However, our morphological and acoustic data does not separate them. This suggests either a case of

cryptic speciation or a higher hidden genetic diversity across a larger geographic range than previously thought. In cases of cryptic species, the real challenge is to accurately identify the species limits (not necessarily their validity) between the involved taxa (*e.g. Silva et al., 2020*). To solve the taxonomic issue of *A. tapajos* and its molecularly divergent lineages, further studies should focus on compare these populations in a more comprehensive framework, exploring different molecular markers, morphological evidence of adults and tadpoles, and acoustic data.

In addition to *Ameerega munduruku Neves et al., 2017*, *Pristimantis pictus Oliveira et al., 2020*, *P. pluvian Oliveira et al., 2020*, and *Proceratophrys korekore Santana et al., 2021*, *Allobates paleci* represents the fifth species of amphibian described for the Teles Pires River region in the last 5 years. These findings reinforce the high levels of hidden diversity in this river basin. Unfortunately, this region has also been severely impacted by anthropic pressure in the last two decades, mainly by the establishment of large hydroelectric power plants and livestock farms (*Fearnside, 2005*; *Fearnside & Pueyo, 2012*). The maintenance of long-term studies and the consideration of this high concentration of type localities in the elaboration of public policies is crucial to preserve this remarkable biodiversity.

## ACKNOWLEDGEMENTS

We are in debt with Adrian A. Garda for the critical review of the manuscript, improvement of the English, and provide his lab so we could do extraction, amplification, and sequencing of DNA used in the present study. We also thanks Albertina Lima who kindly offered several observations and literature suggestions. We would like to thank the referees Leandro Moraes, Diana Rojas, the anonymous 3th reviewer, and the editor Tomas Hrbek for helping to improve the final version of our paper.

### Funding
This work was supported by the Conselho Nacional de Desenvolvimento Científico e Tecnológico (140408/2018-5, 309420/2020-2). The funders had no role in study design, data collection and analysis, decision to publish, or preparation of the manuscript.

### Grant Disclosures
The following grant information was disclosed by the authors:
Conselho Nacional de Desenvolvimento Científico e Tecnológico: 140408/2018-5 and 309420/2020-2.

### Competing Interests
The authors declare that they have no competing interests.

## Author Contributions

- Leandro A. Silva conceived and designed the experiments, performed the experiments, analyzed the data, prepared figures and/or tables, authored or reviewed drafts of the paper, and approved the final draft.
- Ricardo Marques conceived and designed the experiments, performed the experiments, analyzed the data, prepared figures and/or tables, authored or reviewed drafts of the paper, and approved the final draft.
- Henrique Folly performed the experiments, authored or reviewed drafts of the paper, and approved the final draft.
- Diego J. Santana conceived and designed the experiments, performed the experiments, analyzed the data, prepared figures and/or tables, authored or reviewed drafts of the paper, and approved the final draft.

## Animal Ethics

The following information was supplied relating to ethical approvals (*i.e.*, approving body and any reference numbers):

The following information was supplied relating to ethical approvals (*i.e.*, approving body and any reference numbers): Collection permits for this study were issued by ICMBIO (SISBio 79127-1).

## DNA Deposition

The following information was supplied regarding the deposition of DNA sequences:

The sequences are available at GitHub: https://github.com/Rhinella85/Allobates-paleci.

## Data Availability

The final alignment used in the present study and the accession numbers of the sequences used in the present study and genetic distance between the analyzed sequences are available in the Supplemental Files.

## New Species Registration

The following information was supplied regarding the registration of a newly described species:

Publication LSID: urn:lsid:zoobank.org:pub:6B0FDB3B-30B2-471E-9E06-8B7F296F454F.

Allobates paleci LSID: urn:lsid:zoobank.org:act:5DCD25EB-F6F3-4D5C-9B1F-D71669EF634A.

## Supplemental Information

Supplemental information for this article can be found online at http://dx.doi.org/10.7717/peerj.13026#supplemental-information.

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
