# Peer review of "A new Amazonian species of Allobates Zimmermann & Zimmermann, 1988 (Aromobatidae) with a trilled advertisement call"

_PeerJ, doi:10.7717/peerj.13026_

## Round 0.1 · original submission · Major Revisions

Dear Authors,

I received three reviews of your manuscript. All the reviewers were positive, but I agree with the third reviewer that your manuscript needs a major revision.

My principal concerns are:

1) Focus: You are describing a new species (Allobates parecis), but much of the manuscript is focused on the second lineage which shows a certain degree of divergence from one of the lineages identified in the study of Rejaud et al. (2020). This permeates the introduction, discussion, and even enters into the new species description, when those sections should be about the new species they are actually describing.

2) You use genetic distance/divergence to decide that A. parecis is sufficiently different to be described as a new species; you are basing your decision on amount of sequence divergence. You have a lot of samples to carry out a species delimitation analyses, and you use sequence data, than you should/could provide molecular diagnostic characters. Otherwise you do not not need your Bayesian trees.
Figure S1: The bottom is cut off. Also why is your outgroup taxon (Anomalogossus) nested within your ingroup (Allobates)?

L75: use “sacrificed” and not “killed”

L238: “ranging from Brazil”; not sure what you mean here, other species from Brazil?

L264: “these two species” refers to what? A. paleci and A. aff. tapajos? If so, why is A. aff. tapajos being discussed here?

There are other concerns brought up by the reviewers, so please address them during your revision.

For my part, I see a need for focusing on A. parecis and a rigorous analysis of your molecular data which will require a partial rewrite of your manuscript.

Otherwise a nice contribution. I look forward to receiving your revision shortly.

Sincerely,

Tomas Hrbek

·

Basic reporting

Dear authors and editor.

Thank you for the opportunity to review this nice manuscript, which represents a standard description of another new species for the speciose Neotropical genus Allobates. The new species described in this manuscript inhabits southern Amazonia and, based on the results presented, I am fully convinced of the author's hypothesis that this sampled population actually represents a valid and yet undescribed species.

I start my considerations by congratulating the authors for this manuscript, where the effort expended since the gathering of data from fieldwork to the generation and interpretation of results is clearly visible. I consider that this manuscript is sufficiently illustrated, has an adequate structure and a good scientific background, the reference list is adequate and relevant to support this new species description, and raw data are also shared. I highlight the emphasis and the resolution that the authors gave to the acoustic repertoire of the species, something that is less frequent in recent descriptions related to the genus and is very welcome for the knowledge advance in this research line.

Nevertheless, I think that the language, fluidity and standardization of the text can still considerably improve. I made some punctual suggestions in this regard in the revised file that I attach in this review, but as I am not an English native, I did not scrutinize language adjustments. I also suggest a general improvement in figure and table legends, whose level of detail could be greater in some cases, as weel as in the figures of geographic distribution and phylogenetic tree.

Experimental design

No comments, topic already covered in other sections.

Validity of the findings

Following the standard currently required for a robust description of an Allobates species, the authors present quite diverse datasets to answer the question raised about the taxonomic identity of this population. These datasets included molecular, acoustic and morphological variation, which were analysed mostly in a qualitative way (as usual for this type of manuscript). Based on the robustness of the employed methods, I think the manuscript was able to efficiently fill the knowledge gap raised.

Additional comments

I present a quick overview, with some suggestions for the main sections:

Introduction - short but adequate enough and sufficiently presenting the research question.

Methodology - unfortunately, some important data about the species are still missing, such as data about its larval stage and its reproductive behavior, characters that are also very relevant in taxonomy of Allobates. But I understand that such data is more difficult to obtain, and I think the authors did their best to achieve a robust result based on the datasets they had. The reproducibility of the performed methods seem quite ok.

Results - The subsections and content are adequate. One major issue I found is related to the fluidity of the text in the acoustic comparisons section, where comparisons are often truncated and it is very important to clarify the definition of a trilled call in an Allobates species, as this is a very important diagnostic characteristic of the new species. The other major issue was the complete absence of citation and emphasis on a character that I consider very relevant for the diagnosis of this new species (the condition of the dark lateral stripe), but these are all issues that I believe can be quickly corrected, and I included these suggestions in the revised file attached.

Discussion - I believe that all the most relevant topics were sufficiently addressed, I just reinforce again that the fluidity of the text can be improved. Some suggestions are presented in the attached file.

All the best,
Leandro

·

Basic reporting

The paper “ A new Amazonian species of Allobates Zimmermann & Zimmermann, 1988 (Aromobatidae) with a trilled advertisement call pattern” provide a detailed description of a new species of Allobates. Authors describe the new Allobates species based in integrative taxonomic analyses and also reports new morphological and acoustic data for a previously recognized cryptic linage of A. tapajos.

The manuscript is clearly written and provide detail and relevant data and results.

Line specific comments:
L182-192: all species and authors should be present in a consistently way: ex. A. species (authors, year). Also not all species are reported with the respective authors.

L268: a “n” is missing should be “ranging” instead of raging
L282: “ranging” instead of raging

Experimental design

No comment

Validity of the findings

No comment

Reviewer 3 ·

Basic reporting

There are a number of grammatical and tense issues throughout; thorough revision is recommended. Introduction context is confusing and focuses too much on A. tapajos instead of the new species.

The manuscript particularly relies on genetic comparisons to sympatric A. tapajos, but presents none of the most relevant genetic results in the main document. There is no genetic matrix from the Mega analyses, and there is not a phylogenetic tree including this species. As a major part of the argument supporting the new species, some form of these analyses incorporating A. tapajos needs to be brought into the main text. There is also little use in comparing the genetic distance of all to all. In the main text, just compare to close relatives and sympatric/parapatric species.

The tables can be combined into far fewer tables for both the morphology and acoustics.

The 16S sequence for the new species in not in the provided alignment, and no genbank accession number is provided for it. This makes the results not reproducible.

Experimental design

There are no actual statistical analyses of acoustic or morphological data presented. Only the raw data is presented, and is not presented in comparison to other sympatric species. This is wholly insufficient. Statistical analyses comparing the new species to A. tapajos is required to support claims of cryptis.

Validity of the findings

The new species appears to be well justified and described (see statistical analysis note). However, the authors have spent much of the text focussed on a sympatric species, which negatively affects the manuscript.

There is no data presented that supports the authors assertions that the lineage they call A. aff. tapajos is a different species from A. tapajos. The mitochondrial divergences presented in the supplementary phylogeny shows A. tapajos to be monophyletic, and have a similar intra-specific divergence to species such as A. grillisimilis. There are many species in the world with highly geographically structured and deep mtDNA divergences. This alone is not sufficient evidence for the strength of the assertions provided here, particularly given the lack of morphological and acoustic comparisons within A. tapajos. I recommend removing all portions of the text suggesting this is a different species (including assigning aff.), unless further within A. tapajos comparisons are included, with additional geographic data. The case presented is not compelling: the very maximum the authors should suggest that further sampling is needed to assess structure in the species. It’s all a complete distraction from the genuine finding – a new species – and does not improve the paper. This will require a major revision of most sections of the manuscript.

Additional comments

Abstract: The Abstract in general has quite poor grammar and needs improvement. Some errors in tense. The acoustic results should make a statement about statistical support for differences from cryptic sympatric lineage

Line 44 – what is the relevance of the chromosome number statement? Either clarify or remove.

Line 48 – Unclear transition – which group is A. tapajos part of? If the groups are important, then this should be defined. Why is A. tapajos a focus?

Line 51 – lower, not low

Line 65 – why is the cryptic lineage not being described? If it’s because there is not enough data, then remove all references as there is little to no evidence presented to show this to be correct.

Line 75 – euthanised, not killed

Line 78 – Collection, not collect. Also cite your animal ethics approvals.

Line 82 – Put the data in Appendix 1 in the main text. It’s only a couple lines.

Acoustic methods – what sort of analyses were undertaken on the call data?

Morphological methods – again, what sort of analyses were undertaken using the data?

Line 150 – Milli-Q

Genetic methods – How many times was Beast run? It should be run at least 3 times, and convergence of the runs on the same topology assessed using RWTY. The all-to-all genetic distance matrix should not be every individual to every individual, but major lineage to major lineage. But really you should only compare the new species to sympatric species/close relatives, and present that in the main text, for it to useful and interpretable.

Line 215 – familiar – do you mean familial?

Line 217 – Is medium the right word?

Line 235 – for cryptic species, best practice is to identify diagnostic molecular differences following Singhal et al 2018 (Systematic Biology; 10.1093/sysbio/syy026).

Line 237 – “centre of endemism” is not defined so unclear. Delete, or define in the introduction.

Lines 450-458 – remove all information that is not about the species being described.

Lines – 466-468 – “Agrees with” and “often overlaps” is insufficient for a scientific paper. Refer to (and do) statistical analyses.

Lines 472-477 – delete. This is insufficient evidence, and also does not belong in the actual description section of a different species.

Tables 2-6 should be combined into a single table. This would allow easy comparison of the call parameters between the two species, and reduce the space required.

---

## Round 0.2 · Minor Revisions

Dear Authors,

I received a review of your revised MS. You have made major improvements to your manuscript, however, there is still one outstanding issue: your molecular analyses and their interpretation.

You state: “Generic placement. The new species is assigned to the genus Allobates based on molecular evidence (mtDNA 16S)…” However, from your table of distances, you cannot conclude if the new species is Allobates or not. The distances are just a measure of overall similarity of the sequences. If we want classification to reflect evolutionary history of taxa, then genera need to be monophyletic. So you need to have a phylogeny with Allobates and other genera, and see if the new species forms a clade with other species of Allobates.
In the description section you state: “Molecular. Average sequence divergence between the new species and the compared nominal congeners was 16.5%, ranging from 13.2% (A. carajas) to 21.3% (A. niputidea) (Table 4, Supplemental Table S1).” This again just represents a description of overall similarity.

When I was commenting on the previous version of your manuscript, I stated that you did not need your Bayesian tree for what you used the molecular data for (just to generate and report distances). At the end of my comments I also stated that you need to do: “a rigorous analysis of your molecular data”. So I did not mean that you should discard the phylogenetic analyses and just keep a table of distances.

You should provide a phylogeny to show that the new species is Allobates (that it forms a clade with all other species of Allobates), and then you can use this phylogeny to carry out a single locus species delimitation analysis. First you should collapse all identical haplotypes, and then run a maximum likelihood analysis in iQTree. The resulting ML phylogeny will have branch lengths proportional to the number of substitutions, and this ML phylogeny is suitable for analysis in any of the PTP (Poisson Tree Process) family of analyses (PTP, mPTP, bPTP).

Otherwise just reporting distances and using genetic distances to conclude whether or not a particular taxon is distinct from other taxa is really not appropriate.

Please also submit your data to Genbank and include the Genbank accession numbers in your resubmission.

The other sections of your manuscript look fine.

Sincerely,

Tomas Hrbek

Reviewer 3 ·

Basic reporting

The writing is significantly improved in this version. I have provided further specific edits in the manuscript.

For some reason there are no results presented properly. A taxonomy section is not a result - it is an independent section. Actual results should be provided for each section of the methods.

The removal of a phylogenetic tree from the paper is a poor choice, as it is standard in taxonomy these days. The focus on p-distances is extremely unusual, so unusual that I've not seen it in a taxonomy paper in 10 years. Recommend removing the genetic distance and put in a phylogeny. You can do this extremely quickly with IQtree, using the command "iqtree -s alignment.fas -m TEST -st DNA -bb 10000 -alrt 10000 -nt AUTO redo".

The order of the taxonomic section needs revision. It compares to other species before describing paleci. You need to describe it first, so that you can compare.f

Further review of the literature on acoustics, particularly Kohler et al 2017 should be incorporated.

A table of all the individuals and genbank accession numbers is needed - for all individuals in the paper, not just the two primary species. The actual genbank accession numbers need to be put in the paper, not just a statement that they are on genbank.

Experimental design

Not only are the p-distances extremely unusual, but it is very unclear how they were generated. I generated a tree from the alignment file, but many species turn out not to be monophyletic. One example amongst many is the marchesianus is in the phylogeny in three separate places, but it is only represented once in the table. There are also a bunch of what maybe undescribed species, which are in the alignment but no in the distance table. Species such as conspicuus and subfolionidificans are not forming distinct monophyletic clades, but they are presented as separate species in the table.

The alignment is unclear because the individual that I think is the new species is called A. tapajos, and only has a field number. The individual data in appendix one has museum IDs. Remove all field IDs and put in the museum ID so individuals can be traced. Ensure names in the alignment make sense.

Essentially, p-distances hide the uncertainty and are an inappropriate way to present the realities of genetic data for this group. This paper needs a phylogeny, and the p-distances should be removed.

Validity of the findings

Other than the p-distances providing a false sense of a simple genetic result, the other results are good.

Annotated reviews are not available for download in order to protect the identity of reviewers who chose to remain anonymous.

---

## Round 0.3 · accepted · Accept

Dear Authors,

I received a review of your revised manuscript. Thank you for implementing the suggestions of the last revision. Your manuscript is now much improved, and I am happy to accept it.

I have made minor edits (English and style) on the PDF version of your article. Please make these changes either during the page proofs stage or before typesetting; the editorial office should let you know.

Congratulations on a job well done.

Tomas Hrbek